# A Single-Cell Perspective on the Effects of Dopamine in the Regulation of HIV Latency Phenotypes in a Myeloid Cell Model

**DOI:** 10.3390/v17070895

**Published:** 2025-06-25

**Authors:** Liana V. Basova, Wei Ling Lim, Violaine Delorme-Walker, Tera Riley, Kaylin Au, Daniel Siqueira Lima, Marina Lusic, Ronald J. Ellis, Howard S. Fox, Maria Cecilia Garibaldi Marcondes

**Affiliations:** 1San Diego Biomedical Research Institute, San Diego, CA 92121, USA; lbasova@sdbri.org (L.V.B.); alim@sdbri.org (W.L.L.); vdwalker@sdbri.org (V.D.-W.); triley8@asu.edu (T.R.); kau@sdbri.org (K.A.); dlima@sdbri.org (D.S.L.); 2Interdisciplinary Fellowship on neuroAIDS, University of California San Diego, San Diego, CA 92103, USA; 3Heidelberg University Hospital, 69120 Heidelberg, Germany; marina.lusic@med.uni-heidelberg.de; 4Department of Psychiatry, University of California San Diego, San Diego, CA 92121, USA; roellis@health.ucsd.edu; 5Department of Neurological Sciences, University of Nebraska Medical Center, Omaha, NE 68198, USA; hfox@unmc.edu

**Keywords:** HIV, dopamine, latency, single cell

## Abstract

Psychostimulants such as methamphetamine (Meth) induce high dopamine (DA) levels in the brain, which can modify immune cells expressing DA receptors. This is relevant in conditions of infection with the human immunodeficiency virus (HIV), overlapping with substance use. However, the effects of DA on HIV latency phenotypes are largely unknown. We used single-cell methods and gene network computational analysis to understand these relationships, using the U1 latent promonocyte model to identify signatures of latency and its reversal in the context of DA exposure. Our findings point to mechanisms by which high DA levels in the brains of substance users may impact HIV transcription and neuroinflammation. Our data indicate that latency is maintained along with the expression of histone linkers and components of chromatin organization, with increased metabolic pathways that may lead to pathways in neurodegeneration. DA exposure decreased latency signature genes, histone linkers, and protein-containing complex organization components, unleashing inflammatory pathways and HIV gene transcription. Overall, this work suggests that DA can induce latency reversal through mechanisms that can be harnessed to drive cells. The proposed methods developed here in cell lines can be used to identify latency signatures in other HIV infection systems.

## 1. Introduction

Cognitive impairment in HIV and the dysregulation of dopaminergic systems are well-documented by us and others [1,2,3,4,5,6,7,8,9,10,11,12,13,14]. These clinical and physiological disturbances may result from the substance use disorders that are frequent among people living with HIV (PWH) [15,16]. Psychostimulants, such as methamphetamine (Meth) and cocaine, are potent inducers of dopamine (DA) in the central nervous system (CNS). HIV enters the CNS during acute infection and remains latent or replicates at low levels in microglia reservoirs. While DA is fundamentally involved in mechanisms of reward, cells targeted by HIV in the CNS and elsewhere express all five DA receptors [1,11], with phenotypic characteristics that can be deeply affected by the DA-rich brain environment of substance users [1,2,5,7,13,17,18,19,20,21,22,23]. Indeed, many reports indicate that psychostimulants, such as methamphetamine, increase cellular susceptibility to HIV entry and spread [1,2,11,24], which ultimately may be critical for increased viral load in the brains of users, as observed in animal models and human cohorts [25,26]. A question that remains, however, is the mechanistic basis for more frequently detectable viral loads in the brains and cerebrospinal fluid (CSF) of substance users and, in particular, what the role of DA in the increased susceptibility is. Currently, an outcome of effective antiretrovirals is the suppression of HIV in peripheral sites and fluids. However, latently infected long-lived innate immune cells with integrated viral sequences can either reactivate and reseed the infection or maintain a low-level production of HIV proteins with active roles in host cell chronic pathogenesis, even in PWH with viral suppression on ART [27,28]. We have previously shown that DA is a factor involved in cell-to-cell HIV spread [1], but most importantly, in the latency reversal that is detectable in the CSF of methamphetamine users [2]. Here, we report on experiments in a promonocytic cell line latency model [29] conducted to understand how elevated levels of DA found in the brains of methamphetamine users [30] may trigger reactivation of the latent virus from the reservoirs of myeloid origin. The choice of U1 cells as a model was based on the consistency and predictability of the latent state without variables linked to technical factors, with the goal of developing a strategy to facilitate the identification of virally infected/latent cells in situations of lack of reporters, or within the heterogeneity of myeloid target cells in vivo, preventing sorting or recognition.

Our previous studies have indicated that DA reverses latency in a significant fraction of, but not all, latent promonocytes [2], leading us here to probe the use of single-cell transcriptome analysis to compare features triggered by DA in cells that show signs of reversal in comparison to cells that maintain latency. The single-cell strategy enabled the detection of differences in the transcriptional landscape of all nuanced subpopulations showing elevated transcription of HIV gag-pol-env-tat-nef or no detectable viral transcription in response to DA, as well as its signatures, to define the requirements of latency regulation and pathways that may be harnessed for understanding neuroimmune interactions in populations of drug users. The results presented here further confirm the role of DA in latency reversal but also suggest exclusive pathways and orchestrated transcriptional programs that can be explored to control and shift the HIV latent state to active phenotypes.

This study specifically aimed to develop strategies that can help to elucidate the mechanisms by which elevated DA levels trigger the reactivation of HIV from latency in myeloid cells and distinguish cells that remain latent for contrasting the heterogeneity of transcriptional responses of DA-sensitive and DA-insensitive cellular populations. By employing single-cell RNA sequencing to an in vitro myeloid latency model, we seek to identify unique gene expression patterns that may distinguish these cellular responses, thereby contributing novel insights into the neuroimmune interactions applicable to brain-derived targets pertinent to HIV persistence in substance users.

## 2. Materials and Methods

### 2.1. Promocytic Cell Lines and Culture Conditions

Chronically infected HIV-1 promonocytic (U1, ARP-165) cell lines were originally derived by limiting the dilution cloning of U937 surviving an acute infection with HIV-1 [29]. The U1 cells (ARP-165) [29] were obtained from the NIH HIV Reagent Program, Division of AIDS, NIAID, NIH, generously contributed by Dr. Thomas Folks. The parent U937 cells (CRL 1593.2) were obtained from the American Tissue Culture Collection (ATCC, Manassas, VI, USA). Both U1 and U937 cells were cultured in RPMI 1640 containing 2.0 mM of L-glutamine and 10% heat-inactivated fetal bovine serum (Lonza Bioscience, Morrisville, NC, USA), and maintained in the log phase with >98% viability before plating at 10^6^/mL in 12-well plates and stimulation. The cells were used with no more than 5 passages.

### 2.2. Cell Culture Treatments

DA concentrations were optimized before experimentation. Dopamine hydrochloride (H8502, Sigma Aldrich, St. Louis, MO, USA) was used, at 1 and 10 μM, to mimic median range levels found in the brains of drug users [30] and did not produce significant differences in a pilot. The single-cell results that are shown were obtained with 10 μM DA. Recombinant human heterodimer MRP8/14 proteins (S100A8/S100A9, R&D Systems, Minneapolis, MN, USA) were used at 1 μM, as previously described [2].

### 2.3. Single-Cell RNAseq Samples Preparation

Single-cell RNAseq was performed using 10× Genomics Chromium Next GEM Single Cell 3ʹ Kit v3.1 (10× Genomics, Pleasanton, CA, USA) as per the manufacturer’s recommendations. Briefly, upon stimulation, the cells were collected by centrifugation (400× *g*, 5 min, at room temperature), and washed with 1 mL PBS containing 2% bovine serum albumin (BSA, Sigma-Aldrich, St. Louis, MO, USA). We loaded 10,000 fresh cells along with gel beads containing unique barcodes and reagents, on 10× Genomics Chip G, using the 10× Chromium Controller that performs the gel-emulsion (GEM) single-cell droplets mRNA capture process. Within each GEM, cell lysis occurred; mRNA was copied into cDNA by reverse transcription, tagged with a cell barcode and UMI (Unique Molecular Identifier). The GEMs were then broken; cDNA was cleaned up and amplified. Using Chromium single cell 3′ Reagent kits with v3.1 Chemistry Dual Index reagents, the cDNA was subjected to fragmentation and Illumina sequencing-ready library by the addition of sequencing adapters. The size of the libraries as well as the quantities were, respectively, measured by using the Agilent 2200 TapeStation System (Agilent, Santa Clara, CA, USA) and QubitTM 1× dsDNA HS Assay Kit (Invitrogen, Q33231, Waltham, MA, USA) before testing. A Single Cell 3′ Gene Expression library was generated using a 25% of the total extracted cDNA. Samples in duplicates were sequenced on Illumina NovaSeq 6000 with an average depth of at least 20,000 100 bp × 100 bp PE reads per cell. 

### 2.4. Single-Cell RNAseq Data Analysis

(i)Alignment and Quality filtering—Sequencing data processing were performed using the Cloud-based Cell Ranger 7.2.0 platform and visualized in the Loupe 7.0 browser (10× Genomics). Specifically, the scRNA data was demultiplexed and aligned to the customized hg38 Homo sapiens reference genome (NCBI RefSeq assembly GCF_000001405.40) which was combined with a chromosome representing the HIV-1 genome (AF033819). The generated sequencing libraries were then aggregated using the function Cell Ranger Aggr (v3.1.0); low-quality single-cell transcriptomes were filtered based on the UMI count [500 to 30,000], gene count [300 to 5000], and mitochondrial percentage [less than 15%]. Table 1 shows the Cell Ranger-derived quality controls.(ii)Data normalization, batch correction, and clustering analysis. Following data normalization in Seurat v4 [31], batch correction was performed using Harmony. Using the 2000 most variable genes (default parameters), and 50 most significant principal components, we performed linear dimensional reduction and built a neighborhood graph using the t-SNE low-dimension visualization tools in Seurat. Cluster-specific pathways and biological processes were analyzed using iPathwayGuide (AdvaitaBio, Ann Arbor, MI, USA) [32], as well as the Database for Annotation, Visualization and Integrated Discovery (DAVID) [33], and visualized using GeneMania [34] in Cytoscape 3.10.2 [35,36].(iii)Differential gene expression (DGE) analysis was conducted with the DESeq2 R Package [37]. Cell Ranger was integrated to calculate DGEs and generate volcano plots for the visualization of differential gene expression upon different stimulations: non-treated (NT) vs. dopamine (DA), within and between U937 and U1 cells (Table 1). Statistical significance was tested using the Wilcoxon matched-pair signed-rank test. Cell Ranger data were converted to Seurat for aggregation and cluster analysis.(iv)All raw and processed data were deposited in GEO with the assessment number GSE278043.

### 2.5. Systems Biology and Visualizations

Each cluster was treated as an individual experiment for input in .txt format, used to identify significantly impacted signaling pathways, gene ontology terms, disease processes, predicted upstream regulators, and putative mechanisms based on significant gene expression signatures. Comparisons between clusters of interest were performed using meta-report features in iPathwayGuide. Gene network analysis was performed in Cytoscape v3.10.2 [36], with GeneMania plug-in [38,39], and Homo sapiens sources from BioGRID_ORGANISM [40,41,42]. Permutation analysis in iPathwayGuide was used to rank gene lists by cluster matched to the Gene Set Enrichment Analysis (GSEA) database [43,44] using the keyword “HIV”.

## 3. Results

The transcriptional landscape of latently infected U1 cells and their parent uninfected U937 cell line were compared in the absence and presence of DA treatment. The integration of heterogeneous features in the dataset was visualized in a probabilistic framework using uniform manifold approximation and projection (UMAP) for dimensional reduction (Figure 1). DA or vehicle exposure were carried out for 24 h and the cells were prepared for single-cell RNAseq, as described above. We identified a marked heterogeneity within the cell lines at any given condition and shifts due to treatment, detectable in unsupervised analysis both by dimensionality (Figure 1A) and by subset clusters with common and exclusive profiles (Figure 1B,C). The data revealed that the exposure to DA deeply modified gene expression and signatures, both in uninfected U937 and in latent U1 cells (Figure 1A). Dimension reduction visualization of individual and merged conditions indicated cell clusters with exclusive transcriptional profiles resulting from the HIV latent virus (Clusters 2 and 10) or DA (Clusters 1 and 8), as well as their interactions (Clusters 3 and 6), (Figure 1B,C), and clusters with overlapping dimensions across conditions (See Appendix A for a detailed assignment of clusters to experimental conditions). The observation of individual clusters with overlapped single-cell events associated with the transcription of HIV genes (Figure 1C—black dots), indicates an enrichment of HIV gag, pol and env-expressing cells in latent cells. Merged and individual dominant clusters were analyzed for estimating the effects of DA, alone and in the context of latency. Appendix A shows predicted regulators for behaviors in response to latency and DA exposure, as well as genes in each cluster that have been identified in HIV-association studies.

### 3.1. Uninfected, Unstimulated Control Signatures

Clusters 4, 9, and 11 characterized untreated, uninfected U937 parent promonocytes, referred to as controls (Figure 1 and Figure 2). These clusters had a limited number of significant genes analyzed in a rank diagram that indicated overlap in expressed genes, such as the nucleoside diphosphate kinase 2 (NME2) and the ribosomal protein 17 (RPS17), expressed in all three clusters (Figure 2E). The metallothioneins 2A and 1G (MT2A and MT1G), and the amyloid beta precursor protein (APP) were expressed by both dominant Clusters 4 and 9, which had a higher number of significant exclusive signatures compared to Cluster 11 (Figure 2D), assigned to pathways important for physiology, such as mineral absorption, purine and pyrimidine metabolism, metabolic pathways, neurodegeneration and ribosomes, as seen in the circos diagrams (Figure 2A–C).

### 3.2. Signatures of DA Stimulation on Uninfected Control Cells

Clusters 1 and 8 were enriched in the U937 cells’ response to DA alone, but not in U1 cells in conditions of HIV latency (Figure 1B). These clusters overlapped in the dimensionality reduction plots; however, their profile characterization indicated different phenotypes. Cluster 1 was dominant when compared to Cluster 8 due to its higher number of cells, but was phenotypically less active, with a smaller number of genes showing significant expression (Figure 3A,B) compared to Cluster 8 (Figure 3C,D). Cluster 8 had 97 significantly upregulated genes and 12 downregulated genes, while Cluster 1 had four out of six significant signatures overlapping with Cluster 8 (Figure 3F). Thus, despite being less dominant in numbers, Cluster 8 signatures were used to describe the effects of DA alone in U937 cells. The four overlapping signatures between these clusters were two genes downregulated by DA, the nucleoside diphosphate kinase 2 (NME2), the macrophage migration inhibitory factor (MIF), and two upregulated genes, the enzyme cAMP-specific 3′,5′-cyclic phosphodiesterase 4D (PDE4B) and the latent-transforming growth factor beta-binding protein 1 (LTBP1). The two genes exclusively upregulated in Cluster 1 were SLC1A3 (solute carrier family 1 member 3, glutamate transporter) and PLAC8 (placenta-specific 8). Significant signatures in Cluster 8 established networks based on activation, with predicted regulation by TNF and IL4, suggesting the activation of inflammatory signals. The signatures in Clusters 1 and 8 were linked to pathways in morphine addiction and metabolism (Figure 3A,C). Common biological processes included response to stimulus, organismal regulation, chemotaxis, and hormonal synthesis and response pathways (Figure 3B,D). Cluster 8 was linked to HIV-1 infection, endocytosis and apoptosis pathways, and biological processes linked to signal transduction and immune system-related responses (Figure 3D). The residual expression of Clusters 1 and 8 in U1 cells (Figure 3G) was correlated with HIV transcriptional activity, as indicated by the dark dots in the cluster representations (Figure 3A,D).

### 3.3. Signatures Restricted to Unstimulated HIV Latently Infected Cells

Clusters 2 and 10 were unique to U1 latent cells and were eliminated by DA stimulation (Figure 4K), suggesting that these clusters may include critical latency signatures shifted by DA in correlation with HIV transcriptional reactivation (Figure 4). Cluster 2 was dominant, both in the number of events and due to a higher number of significantly expressed genes with significant overrepresentation in pathways with strong gene interactions, indicating orchestrated cellular phenotypes. For instance, Cluster 2 phenotypes included genes involved in the integrity of ribosomes (GO:0003735, *p* = 2.8 × 10^−23^), with 41 out 176 significant signatures that were upregulated ribosome-related structural components (Figure 4A), predicted to be regulated by the limiting rate eukaryotic initiation factor 5 (EIF5) (Figure 4B,H). Other significant pathways forming strong gene networks were annotated to neurodegeneration (Figure 4C), metabolic pathways (Figure 4D), thermogenesis (Figure 4E), neutrophil extracellular trap formation (Figure 4F) and oxidative phosphorylation (Figure 4G), with overlapping and unique signatures and kinases. The neutrophil extracellular trap formation pathway (Figure 4F) contained integrin ITGAL, voltage-gated channels in mitochondria, and underlying inflammation. These networks indicate the orchestrated behavior of genes interacting via activation, physical binding, and reactive and catalytic binding (A, B, R and C connectors, respectively). Molecular processes assigned to Cluster 2 indicated that several histone linkers acting on chromatin organization were significantly upregulated. Cluster 10, which also characterized U1 latent cells, had 20 significant genes, 18 downregulated and 2 upregulated, all of them overlapping with Cluster 2 (Figure 4J), but not establishing gene network interactions. Upregulated genes in Cluster 10 were the TLR adaptor interacting with endo-lysosomal SLC15A4 (TASL) and proteinase 3 (PRTN3). Moreover, the genes in Cluster 10 did not establish any pathway-based interactions.

In addition to EIF5 (Figure 4B), the three-prime repair exonuclease (MTREX) encoding a protein that removes damaged nucleotides from DNA molecules was also a predicted regulator of latency phenotypes identified in Cluster 2 (Appendix A), while predicted regulators of signatures in Cluster 10 were colony stimulating factor 3 (CSF3) and STAT4 (Appendix A). Both clusters showed representation of signatures that have been observed in gene set representation analyses (GSRA) linked to HIV-1 Gag Pol Nef in immunized humans, and detectable in PBMCs [45] (M40868, Appendix A).

### 3.4. Signatures of DA Stimulation on HIV Latent Cells

The stimulation of U1 latent cells with DA caused enrichment of Clusters 3 and 6 (Figure 5), strongly associated with the activation of myeloid functions, including phagocytosis and antigen presentation, as well as pathways linked to viral pathogenesis. Cluster 3 was dominant, due to the higher number of events and due to the higher number of significant signatures annotated to pathways and molecular processes (Figure 5A and Figure 5B, respectively) and establishing pathway-based gene network interactions indicative of orchestrated responses (Figure 5C,D). These gene network interactions were annotated to chromosome condensation and remodeling (Figure 5C), and to stress, defense and immune responses (Figure 5D). Overall, the dominant Cluster 3 representing the phenotype of DA-induced latency reversal, was characterized by the downregulation of histone linkers and elements of chromatin condensation and remodeling, as well as components of the ribosome structure, assigned to DNA-sensing mechanisms, splicing, and pathogenesis. Molecular mechanisms of stress and immune defense included adhesion molecules, such as ITGAL and S100A4, and activation markers such as the allograph inflammatory Factor 1 (AIF1), also known as Iba1, the tissue inhibitor of metalloproteinases 1 (TIMP1), and the apoptosis-associated speck-like protein containing a CARD (PYCARD), indicating a contribution of the inflammasome.

Only five signatures were present in Cluster 6; all of these overlapped with Cluster 3 (Figure 5G). These overlapping signatures were the upregulation of two genes: the long non-coding RNA leucin-rich repeat containing 75A (LRRC75A) and resistin (RETN), linked to the positive regulation of viral replication (Figure 5F). the overlapping signatures included the downregulation of three genes: the FK506 binding protein 5 (FKBP5); the serine-arginine protein kinase 1 (SRPK1); and the brain-expressed X-linked 1 (BEX1), involved in the spliceosome (Figure 5F).

### 3.5. Signatures of Latency Reversal Enhanced by DA Stimulation and Their Regulatory Relationships Defined by Gene Network Analysis

The transcription of HIV-1 genes in U1 cells was significantly enhanced by DA exposure (Appendix A). The effect of DA on U1 cells included an increase in the production of viral proteins, as indicated by intracellular and released p24 levels (Appendix A). Computationally, U1 cells with positive transcription of HIV genes Gag-Pol-Env were sorted and their single-cell transcriptional profiles were compared to U1 cells negative to HIV genes regardless of DA (Figure 6).

Signatures of latency reversal included 170 significantly upregulated and 15 downregulated genes. Appendix A shows the fifty most upregulated ones, and Appendix A shows all the 15 that downregulated ones in cellular events with positive HIV Gag-Pol-Env gene transcription upon comparison to latent cells negative to the expression of the same genes. HIV genes Gag, Pol, Vif, Tat, Env, Vpu, Vpr, Nef and Rev were the most upregulated genes by DA in positive cells (Appendix A). The examination of the gene list in the Database of Annotation, Visualization and Integrated Discovery (DAVID) resulted in assignments to inflammatory response and immunity (*p* = 1.1 × 10^−12^), chemotaxis and cell adhesion (*p* = 1 × 10^−6^), and regulation of transcription (*p* = 1.9 × 10^−2^) as significant biological processes in the phenotypes of active HIV transcription. These phenotypes were also identified in iPathwayGuides, as shown in Figure 7, with predicted regulators, such as TP53, TNF, IL10 and IL4/IL13, involved in both the upregulation and downregulation of genes (Figure 7C,D). Additional regulators that were identified included lamin B1 (LMNB1) and sirtuin-1 (SIRT1) (Appendix A). Supporting the notion that DA is a trigger of the changes linked to latency reversal, neuroactive ligand–receptor interaction was a major pathway identified in these cells, followed by immune and viral pathways (Figure 7A). Biological processes were centered in immune activation, and protein-containing complex organization, where chromatin structure and spliceosome components are highly represented (Figure 7B).

For a prediction of HIV latency mechanisms disrupted by DA in the U1 latent model, we compared signatures in dominant clusters characterizing unstimulated latent cells (Cluster 2) vs. DA-stimulated latent cells with detectable enhanced viral transcription (Cluster 3). The contrasting patterns in gene networks containing ribosomal components of the spliceosome (Figure 8A,B, *p* = 3.9 × 10^58^), and proteasome (Figure 8C,D, *p* = 4.4 × 10^8^), visualized by colors, suggest that DA may relax critical mechanisms that explain a decrease in protein–RNA complex organization in protein-containing complexes affecting the regulation of inflammation (Figure 7B) and simultaneously allowing for the transcription of viral genes.

The introduction of neighborhood filters using JActiveModules applied to all the identified signatures of latency reversal (170 upregulated genes and 15 downregulated genes, see Appendix A) indicated subnetworks with 4 representative patterns (Figure 9) to indicate contrasting and orchestrated expression that distinguished Cluster 2 (dominant latent phenotype) from Cluster 3 (dominant latency reversal phenotype), where connectors indicated pathway (blue lines) and physical interactions (red lines).

In Pattern 1, connected genes were significantly downregulated in latent cells (Figure 9A), but showed contrasting expression levels or no significant changes in cluster cells with positive viral transcription (Figure 9B). Pattern 1 contained genes involved in biological processes of host–virus interactions (Benjamini *p* = 1.8 × 10^−7^), apoptosis (*p* = 0.001), phagocytosis (*p* = 0.009), the inclusion of genes with kinase properties (*p* = 8× 10^−4^) and common promoter binding motifs to TET2 and SPI1, strongly annotated to pathways of trans-endothelial transmigration (*p* = 5.2 × 10^−4^).

Pattern 2, on the other hand, was characterized by genes that were significantly overrepresented in latent cells but decreased or no longer significant in cells positive to HIV gene transcription (Figure 9C,D). Pattern 2 was linked to biological processes of translation regulation (*p* = 0.05), with molecular functions of ribonucleoprotein (*p* = 9.4 × 10^−19^) and annotated to the ribosome (*p* = 5 × 10^−19^) and viral infection pathways (*p* = 1.9 × 10^−17^).

Pattern 3 was a network of genes downregulated in cells with positive HIV transcription following DA, some of which showed diverse or opposite expression levels in latent cells (Figure 9E,F). This subnetwork pattern contained genes also involved in host–virus interactions (*p* = 0.0007), translation regulation (*p* = 0.008) and RNA processing (*p* = 0.02). These genes were linked to molecular functions of helicase (*p* = 0.005), RNA binding (*p* = 0.005), chaperone (*p* = 0.007) and DNA binding (*p* = 0.05); however, links with pathways were not significant. Importantly, Patterns 3 and 2 showed substantial overlap in gene representation, which included histone linkers H1-3, H1-4, and H1-5, significantly downregulated in DA-treated cells with positive viral transcription (see also Appendix A and Appendix A for protein validation), and with connectors to H1-2 and other histone modifiers in DA-induced Pattern 3 (Figure 9E,F). This analysis further suggested the involvement of nuclear components in latency and its reversal caused by DA.

Pattern 4 was characterized by genes mostly significantly upregulated in cells with positive HIV gene transcription, but with inverse or attenuated levels in cells that remained latent (Figure 9G,H). The genes in this network are annotated to primary mitochondrial disease (*p* = 0.0004). Biological processes were associated with innate immunity and inflammation (*p* = 0.003), respiratory chain (*p* = 0.008), and molecular functions of oxidoreductase (*p* = 0.04).

### 3.6. Summary of Results

Overall, the computational strategies and gene network analysis tools applied here to a latent innate immune cell model allowed for the identification of interactions linked to latency or its reversal in the context of DA exposure, with annotations that can explain the mechanisms by which DA may affect HIV latent cells in substance users, as summarized in a schematic cartoon in Figure 10.

## 4. Discussion

The goals of this study were to enhance our understanding of HIV latency relative to active transcription in myeloid cells and to explore the effects of external factors, particularly dopamine (DA), on transcriptional regulation. Reports of increased viral load in the brains of stimulant substance users suggest that HIV latency mechanisms can be disrupted by complex factors in these individuals [46]. Given the well-known relationship between psychostimulants and DA, our experimental design examined the transcriptional profiles of control and latently infected promonocytes to uncover DA-induced gene expression changes that could explain latency reversal in substance users. We previously demonstrated that DA promoted HIV entry into innate immune cells [1] and enhanced viral transcription in a latency model [2]. However, DA activates HIV gene transcription in only a subset of cells. Single-cell RNA sequencing enabled us to isolate these responsive subpopulations and compare them to uninfected, latently infected, and DA-stimulated cells within a promonocyte cell line system. This approach lays the groundwork for future in vivo studies.

A series of analytical approaches proved useful for identifying cellular clusters expressing specific behaviors as a result of DA stimulation and in the context of HIV. A strength of the introduction of systems strategies is the detection of biological processes rather than single gene signatures, increasing the power of discovery of redundancies when response patterns are characterized by more than one cluster. For instance, Clusters 4, 9, and 11 characterized unstimulated U937; Clusters 1 and 13 characterized U937 cells stimulated with DA; Clusters 2 and 10 characterized unstimulated U1 latent cells; and Clusters 3, 6, and 12 were found in U1 latent cells stimulated with DA. The separation in clusters results from differences in signatures ranked by dominance. Although common signatures linked to the overall cluster behaviors could be identified in some cases, systems interactions and annotated pathway and biological process differences were more useful to understand the role of DA as a factor reversing latency.

The significance of our approach is enhanced by the lack of an overall understanding of cellular factors and cellular pathways mediating HIV latency and reactivation. In the U1 cell model used in this study, an attenuated Tat confers the characteristic of latency [29,47]. Yet, PKC activators and latency reversal agents strongly increase p24 in these cells’ supernatant because of intact and viable HIV replication in their totality [2,48], which ensures that the model is valid for developing a study pipeline. Indeed, we were able to detect patterns that characterize latency and HIV transcription in the context of DA exposure, with signatures that provided mechanistic insights.

The uninfected parent promonocytic cells displayed a limited number of distinctive transcriptional signatures, and notably increased expression of nucleoside diphosphate kinase B (NME2), which facilitates the synthesis of nucleoside triphosphates other than ATP, and ribosomal protein S17 (RPS17), as well as the upregulation of amyloid beta precursor protein (APP) and the metallothioneins MT1G and MT2A. These cysteine-rich metallothioneins play key roles in DNA damage repair through metal detoxification, the modulation of mineral absorption, and the enhancement of cellular resistance to apoptosis. Consistent with our observations, previous studies have highlighted the essential function of these molecules in preserving normal monocyte activity [49]. NME2 is a gene encoding a protein complex that helps with trafficking between the endoplasmic reticulum and Golgi and that may have epigenetic functions depending on its tertiary structure [50]. DA stimulation on uninfected conditions, on the other hand, caused the downregulation of NME2 and MIF and the upregulation of PDE4B and LTBP1. The contrasting effect on NME2 expression in control cells following DA indicated a potential signature of DA stimulation in promonocytes.

The dominant phenotype in unstimulated U1 cells characterizing latency involved the high expression of genes encoding ribosome structural elements, as well as replication-dependent histones such as H2AC17 and histone linkers. These findings indicated the contribution of chromatin structure in transcriptional silencing, while indicating that protein-containing complex organization, with a predicted regulatory role of the eukaryotic translation initiation element EIF5, are linked to HIV latency. Interestingly, the signatures of latency were assigned to neurodegeneration, metabolic pathways, oxidative phosphorylation networks, and the presence of molecules controlling leukocyte transmigration, such as ITGAL, which indicate that, despite the latent phenotype, these cells may have the capacity to influence the environment to enhance inflammation. This could be a result of residual viral replication [51] and could explain the persistence of low levels of neuroinflammation in ART-suppressed PWH.

The effects of DA on HIV latent cells were detectable in Cluster 3, where myeloid and inflammatory pathways were significantly enhanced, including PYCARD, an adaptor component of the inflammasome previously reported to be activated by DA in monocytes [52]. Importantly, genes encoding histone linkers and chromatin structural elements were significantly downregulated by DA in latent cells. The computational sorting of cells positively transcribing HIV genes indicated that a decrease in histone linkers is a strong signature of DA-induced latency reversal, suggesting effects on chromatin organization. Moreover, genes linked to stress response, and immune defense, including anti-viral response and inflammation, were enhanced in these cells. The predicted regulators of latency reversal phenotypes by DA included TP53, TNF, as well as IL10 and the IL13/IL4 axis. IL13 and IL4 can control myeloid phenotypes towards suppressor M2 responses [53,54,55]. Overall, DA may shift the response of latent myeloid cells towards activation, with the contribution of changes of chromatin structure, due to the decrease in histone linkers H1-5, H1-4, opening chromatin (see Appendix A) and unrepressing HIV transcription along with pro-inflammatory genes. While others have suggested that DA is a suppressor of inflammation [54,55], our previous studies in other innate immune cell models in the context of HIV infection, as well as in U1 cells containing the latent virus, have indicated that DA can activate a significant subset of cells in a dopamine receptor signaling-dependent manner [1].

An additional strategy involved gene network analysis for identifying genes with relationships based on pathway and physical interactions and showing orchestrated behaviors and expression patterns that could distinguish between latent and reversed HIV transcription in the context of DA exposure. The overexpression of ribosomal and chromatin structure components remained as signatures of interest in latent cells, while latency reversal was linked to inflammation and mitochondrial respiratory chain. The gene networks with connectors based exclusively on pathways were annotated to ribosomal control of transcription and the proteasome, overall assigned to protein-containing complex organization. When physical interactions were added, biological processes were expanded to viral–host interactions, mitochondrial health, and viral infection, supporting the notion that cells with transcriptionally active HIV expression have an active phenotype [56]. Genes annotated to dopaminergic signaling were also identified as perturbed by DA in correlation with increased HIV transcription, further indicating specific DA signaling involved in the activation of these cells.

Additional considerations may be introduced regarding the effects of DA on epigenetic silencers, such as SIRT1, which was predicted as a regulator of DA-induced phenotypes and plays a role in the stabilization of histone linkers. We previously described a loss of Sirt1 and its epigenetic silencing properties as a cause of neuroinflammation in microglia using the non-human primate model of neuroHIV [57]. The results presented here suggest that chromatin structural factors may underlie reactivation and may be considered for intervention approaches.

While the U1 promonocytic cell line is a well-established model for studying HIV latency, it is important to acknowledge the intrinsic limitations associated with the use of immortalized cell lines. The U1 cells, derived from the U937 monocyte lineage, may not fully represent the phenotypic diversity found in primary myeloid cells within the central nervous system of HIV-infected individuals. Furthermore, the cell culture environment lacks the complexity of tissue architecture and systemic influences present in vivo. However, despite the identification of two mutations in Tat causing a partial deficiency that is regarded as a driver of latency [58], baseline p24 was detectable (Appendix A). U1 cells can be induced to produce virus, which in this study was confirmed in the supernatants of the analyzed cultures by p24 ELISA as well as by intracellular cytokine staining. DA increased the percentage of p24+ cells from a 9% baseline to up to 25%, further validating the model for studying the requirements of HIV latency and reactivation. Apoptotic pathways mediated by p53 expression in U1 cells and ectopic Tat can also reactivate HIV-1 transcription and replication from its latent state [59]. However, this model has been also used to examine the mechanisms of Tat-independent transcription [59]. Similarly to cells in vivo, cytokines, such as interferon-gamma (IFN-γ) [60], can also activate HIV expression in U1 cells, including redirecting the production of virions to intracytoplasmic vacuoles [60]. The HIV produced by U1 cells cannot infect new cells, despite producing p24 [61]. Yet, whether the resulting virus is infectious or not does not decrease the value of the model for predicting the mechanisms of reactivation and the phenotypes that allow for the identification of latent states. Other models should be used to investigate the effects of DA on infection, as previously published by us [1,2]. U1 cells as a HIV latent model are useful to our approach, which was aimed at making predictions of mechanisms and especially unique signatures to help to identify latent cells in systems where the virus is not tagged by a reporter or within the heterogeneity of target cells in vivo, and to search for the best strategies to reverse latency by harnessing the immune–dopaminergic response. The effects of DA enhancing viral transcription are replicated in other cellular models of HIV latency that are under investigation, including in primary human microglia and in the C20 microglia cell line (Appendix A). Most importantly, the model provided the development of an analytical strategy guideline that is applicable to other models. The mechanistic insights and the method have utility in validation, confirmation, and for expanding the knowledge gained here. We have confirmed that DA activates myeloid cells.

The single-cell approach allowed us to observe that the effects are heterogeneous, even in cell lines, with a significant subset of responding cells detectable in clusters with viral transcription. U1 cells may be intrinsically heterogeneous due to the replication competence of the integrated virus causing dynamism in integration sites [62], which could potentially explain the phenotypic clusters and response pattern diversity by condition. However, heterogeneity in cell lines has been previously detected using single-cell strategies [63]. Moreover, uninfected parent U937 cells also showed heterogeneity, indicating that the proviral integration sites are not solely responsible for clusters defined by differences in transcriptional signatures. It is important to acknowledge that there are no perfect models; however, the within-cell line diversity is a strength with regard to emulating the diversity of HIV innate immune target cells in the brain, including microglia [64,65,66]. Future studies employing primary human myeloid cells or animal models are warranted to validate the findings reported here, especially in relation to microglia, and to ascertain their relevance in a physiological context.

Whether the reactivation of HIV genes is part of a general effect on gene transcription leading to inflammation caused by DA, or may occur through other means regardless of DA, remains to be confirmed. The decrease in histone linkers in correlation with the active transcription of HIV genes indicates a critical role for DA in nucleosome components and organization, underlying changes in the epigenetic landscape in the context of substance use, as described by us and others [57,67,68]. Overall, the strategy allowed for the identification of a subset of cells responding to DA and their signatures linked to HIV gene transcription, which can be applied to more complex model systems and brains from PWH who are substance users.

## 5. Conclusions

Our findings implicate DA as a key modulator of HIV latency within myeloid cells, potentially offering novel therapeutic targets to PWH that are substance users. Targeting dopaminergic signaling pathways could provide a means to influence latency dynamics, either by promoting latency reversal for viral eradication or by enhancing latency to prevent viral reactivation. Agents that can modulate DA receptor activity without invoking adverse inflammatory consequences could represent a novel class of latency-modifying agents. However, further work in animal models and primary cells is required to explore these therapeutic possibilities in greater detail and to assess their safety profiles.

## Figures and Tables

**Figure 1 viruses-17-00895-f001:**
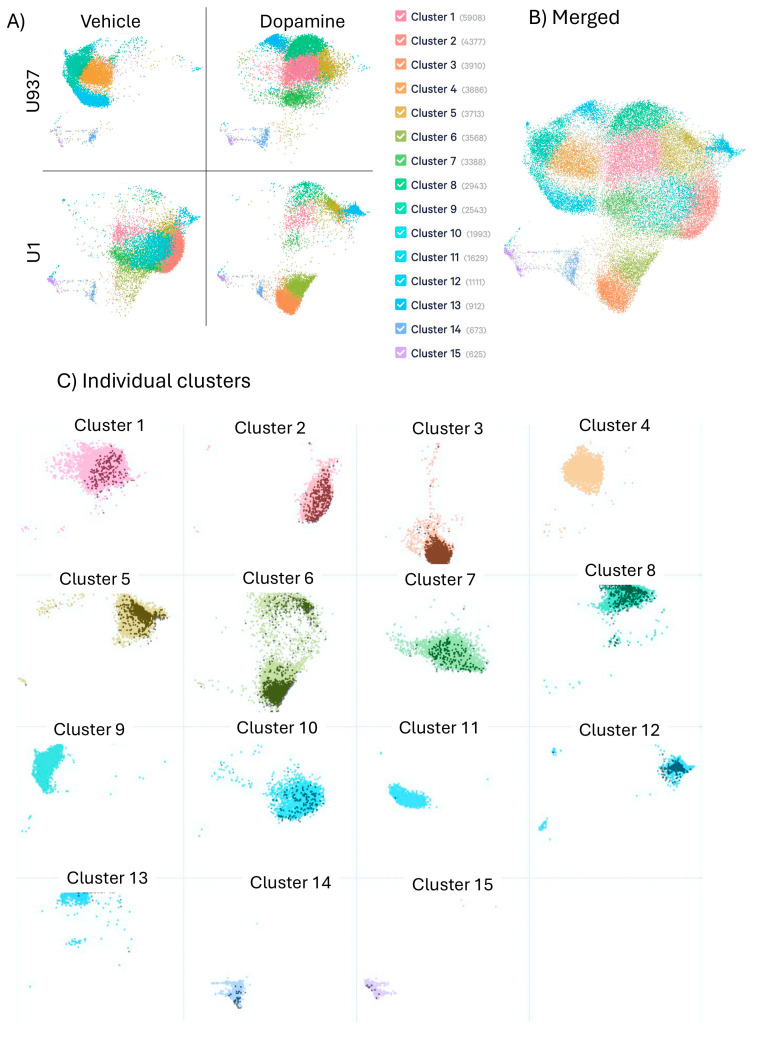
UMAP plots of transcriptomic data in single-cell RNAseq datasets. U937 and U1 cells were cultured with vehicle or 1uM DA for 24 h. Cells were collected and processed for characterization using single-cell RNAseq. Cluster analysis and visualization were performed: (**A**) UMAP for dimension reduction of data in individual treatment conditions; (**B**) merged treatment conditions with color-coded clusters; and (**C**) individual clusters with dark dots indicating events positive to HIV Gag-Pol-Env genes in U1 cell populations.

**Figure 2 viruses-17-00895-f002:**
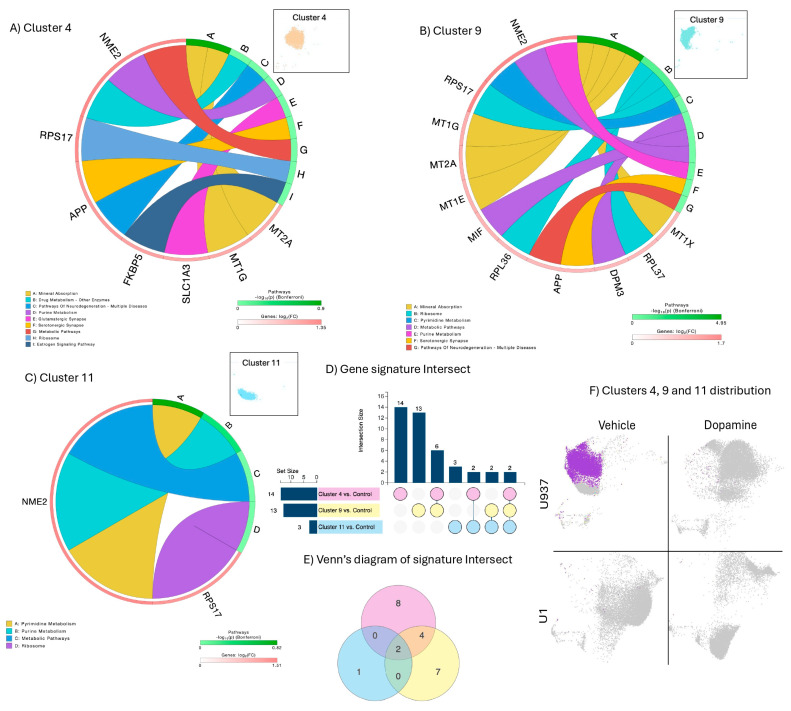
Clusters 4, 9 and 11 characterizations, overlapping signatures, and rank diagram: (**A**) circos plot showing significant genes in Cluster 4 as per Bonferroni-adjusted *p* values and pathway assignment visualization; (**B**) circos plot showing significant genes in Cluster 9 as per Bonferroni-adjusted *p* values and pathway assignment visualization; (**C**) circos plot showing significant genes in Cluster 11 as per Bonferroni-adjusted *p* values and pathway assignment visualization; (**D**) gene intersection analysis between Clusters 4, 9 and 11; and (**E**) Venn diagram indicating signature overlap. (**F**) Clusters 4, 9 and 11 distribution. Merged clusters are highlighted in purple color.

**Figure 3 viruses-17-00895-f003:**
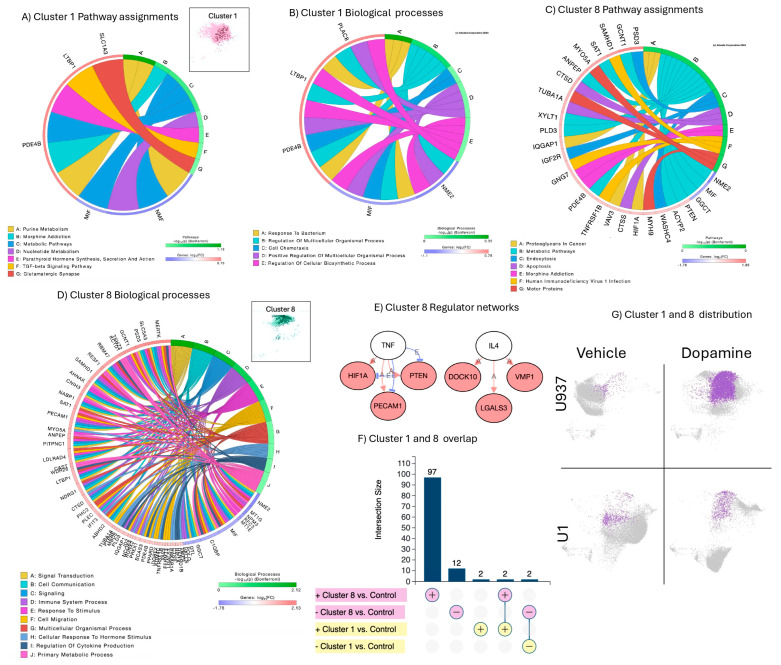
Clusters 1 and 8 characterization, pathway and molecular process assignments, and linked gene network analysis: (**A**) window showing Cluster 1 and a circos plot showing significant genes in Cluster 1, as per Bonferroni-adjusted *p* values and pathway assignment visualization; (**B**) circos plot showing significant molecular processes in Cluster 1; (**C**) circos plot showing significant genes in Cluster 8 as per Bonferroni-adjusted *p* values and pathway assignment visualization; (**D**) window showing Cluster 8 and circos plot showing significant biological processes; (**E**) networks established by genes assigned to significant pathways in Cluster 8, with connectors indicating interactions based on activation, and predicted regulators. Pink nodes indicate upregulated genes, and blue nodes indicate downregulated genes; (**F**) Cluster 1 and 8 directional signature overlap; and (**G**) Cluster 1 and 8 were enriched by DA alone in U937 uninfected parent cells but occurred in small numbers in U1 latent cells.

**Figure 4 viruses-17-00895-f004:**
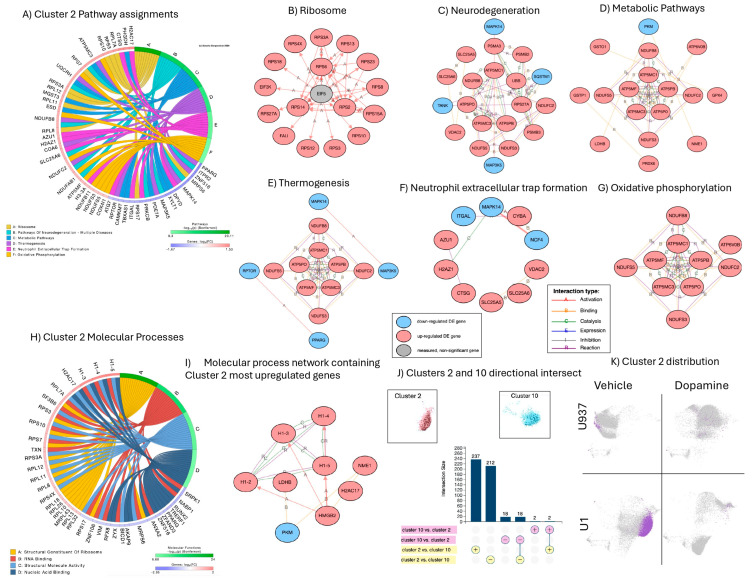
Cluster 2 characterization, assignments, and gene network analysis: (**A**) circos plot showing significant genes in Cluster 2 as per Bonferroni-adjusted *p* values and pathway assignment visualization; (**B**–**G**) networks established by genes assigned to each significant pathway in Cluster 2, with connectors indicating interactions based on activation, binding, catalysis, inhibition, and reaction. Pink nodes indicate genes with increased fold change (FC), and blue nodes indicate decreased FC in relation to average expression; (**B**) ribosome pathway gene interactions; (**C**) neurodegeneration; (**D**) metabolic pathways; (**E**) thermogenesis; (**F**) neutrophil extracellular trap formation pathway; (**G**) oxidative phosphorylation; (**H**) circos plot showing significant molecular processes in Cluster 2; (**I**) gene network analysis for non-pathway overlapping genes assigned to structural molecular activity and nucleic acid binding; (**J**) Clusters 2 and 10 signature overlap, and representation of dark dots indicating HIV Gag-Pol-Env-positive events occurring in U1 cells, and their directional phenotypic intersect, indicating dominant Cluster 2 patterns; and (**K**) map of merged Clusters 2 and 10 unique to U1 latent cells and their pattern suppression in response to DA.

**Figure 5 viruses-17-00895-f005:**
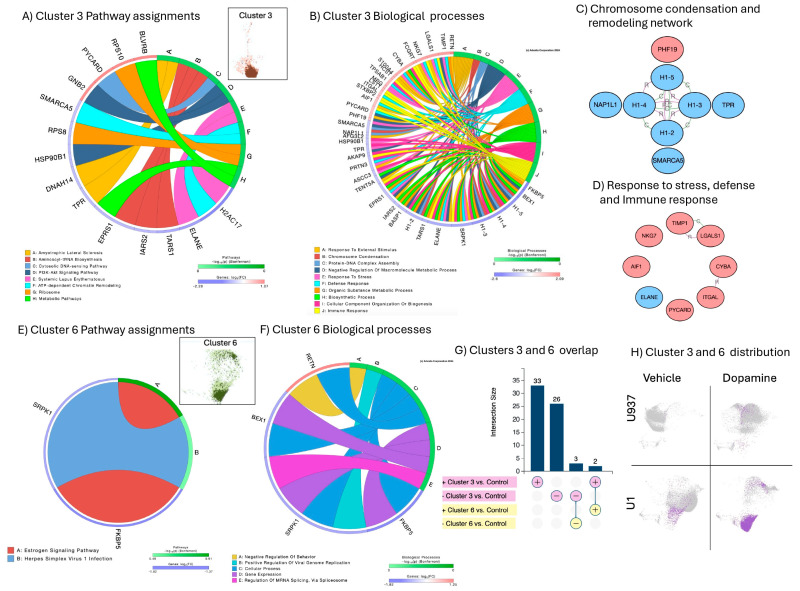
Cluster 3 and 6 pathway and molecular process assignments and linked gene network analysis: (**A**) circos plot showing significant genes in Cluster 3 as per Bonferroni-adjusted *p* values and pathway assignment visualization; (**B**) circos plot showing biological processes assignments in Cluster 3; (**C**) gene network containing genes overlapping between chromatin condensation and remodeling biological processes; (**D**) gene network containing overlapping genes in response to stress, defense, and immune response processes. Connectors indicate interactions based on activation, binding, catalysis, inhibition and reaction. Pink nodes indicate genes with increased FC, and blue nodes indicate decreased FC in relation to average expression; (**E**) circos plot showing Cluster 6 pathway assignments for significant genes; (**F**) circos plot showing Cluster 6 biological processes; (**G**) Clusters 3 and 6 directional intersect, indicating a dominant Cluster 3 pattern; and (**H**) Clusters 3 and 6 distributions unique to U1 latent cells stimulated by DA.

**Figure 6 viruses-17-00895-f006:**
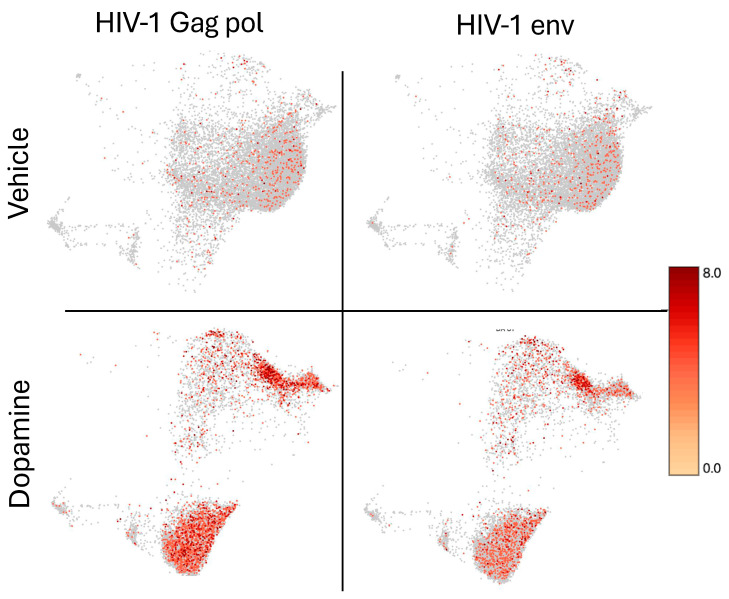
Distribution of HIV gene transcripts in U1 cells at baseline and following DA stimulation for 24 h. The colored scale bar denotes the relative expression values of HIV-1 genes Gag-Pol (left panels) and Env (right panels), at vehicle baseline conditions (upper panels) and after the treatment with DA (lower panels).

**Figure 7 viruses-17-00895-f007:**
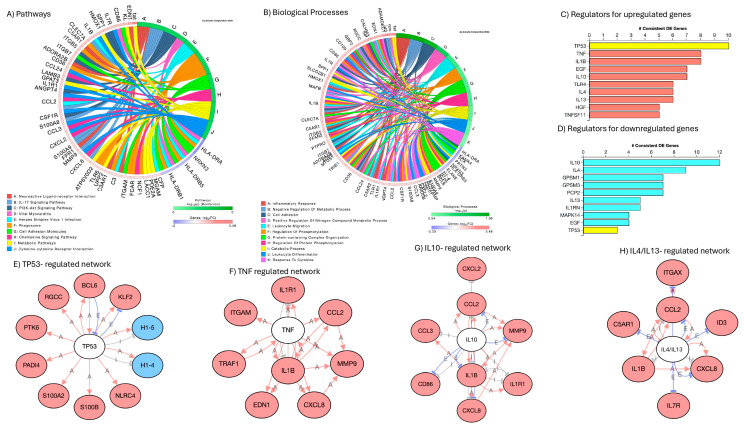
Pathway and biological processes annotations and predicted regulators to signatures that differ between U1 cells with active replication following DA stimulation vs. cells that preserve latency. The comparison between cells transcribing HIV genes and cells that do not result in significantly increased and decreased gene signatures: (**A**) circos plot indicating main annotated pathways; (**B**) circos plot indicating main biological processes linked to gene signatures; (**C**) predicted regulators of upregulated genes; (**D**) predicted regulators of downregulated genes; (**E**–**H**) gene network interactions with consensus predicted regulators, where white nodes represent predicted regulators, pink nodes represent genes upregulated, and blue nodes indicate genes downregulated in correlation with active HIV transcription; (**E**) TP53-regulated gene signatures; (**F**) TNF-regulated genes; (**G**) IL10-regulated genes; and (**H**) IL4/IL13 regulated genes.

**Figure 8 viruses-17-00895-f008:**
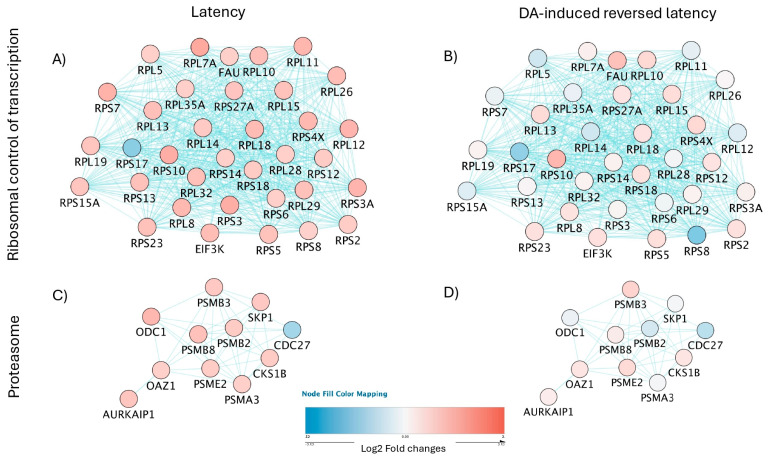
Pathway-based connections among gene signatures significantly perturbed by DA in U1 latent cells. Pathway-based gene networks identified in: (**A**,**C**) Clusters 2 and 10 that characterize U1 latent cells, in comparison to (**B**,**D**) Clusters 3 and 6 that characterized DA-treated cells exhibiting latency reversal. The most significant pathway annotations were (**A**,**B**) ribosomal control of transcription (Benjamini, *p* = 3.9 × 10^−58^) and (**C**,**D**) proteasome (Benjamini, *p* = 4.4 × 10^−8^). Colors indicate upregulation (shades of red) and downregulation (shades of blue). Blue connectors indicate links through pathway.

**Figure 9 viruses-17-00895-f009:**
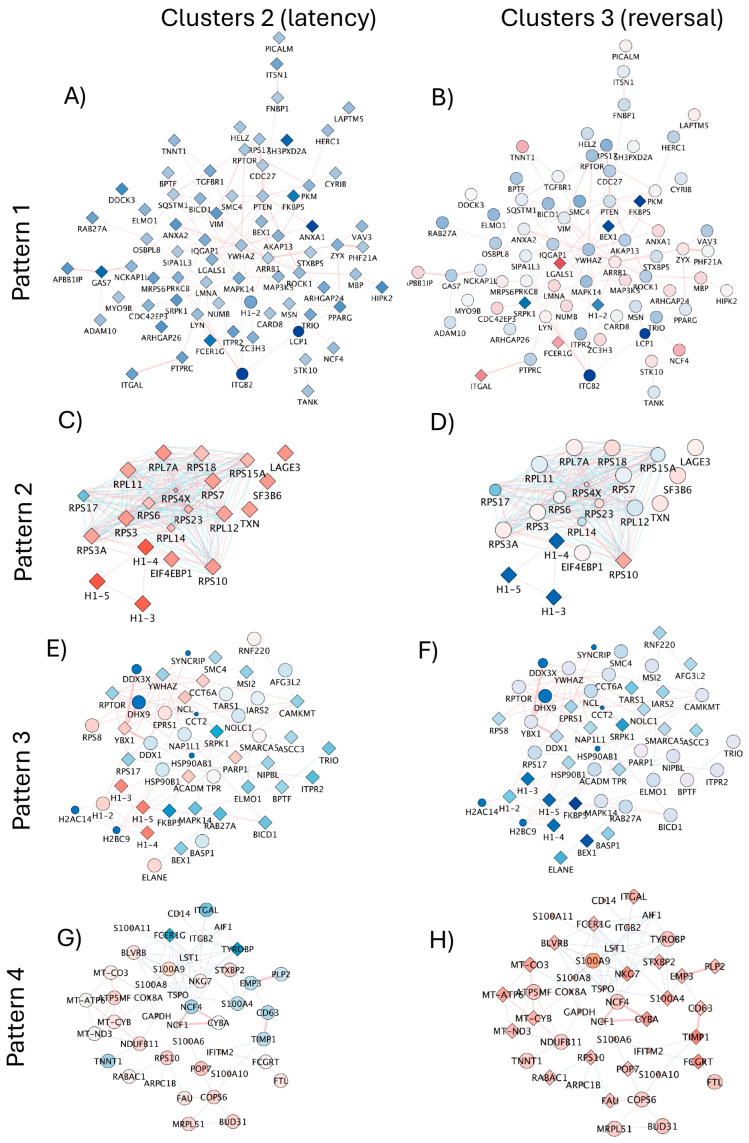
Subnetworks with orchestrated expression patterns and contrasting differences between cluster sets representing latent cells in resting state and reversed latency following DA stimulation. Subnetworks were identified in the merged pathway and physical interactions by applying filters using JActiveModules in Cytoscape 3.10.2. Subnetworks indicated 4 expression patterns that allowed a parallel visualization of contrasting behaviors between dominant response clusters. Cluster 2, representing the dominant latency phenotype (**A**,**C**,**E**,**G**), was visualized in parallel to Cluster 3 (**B**,**D**,**F**,**H**), representing the dominant DA-induced latency reversal phenotype. The identified modules reflected: (**A**,**B**) Pattern 1 containing 72 nodes; (**C**,**D**) Pattern 2 containing 21 nodes; (**E**,**F**) Pattern 3 containing 48 nodes; and (**G**,**H**) Pattern 4 with 59 nodes. Red connector lines indicate physical interactions. Blue connectors indicate pathway interactions. Transcriptional expression levels are indicated by node colors, with shades of blue indicating downregulated and shades of red indicating upregulated genes. Diamond shapes indicate statistical significance (adjusted *p* < 0.05), and circles indicate non-significant genes.

**Figure 10 viruses-17-00895-f010:**
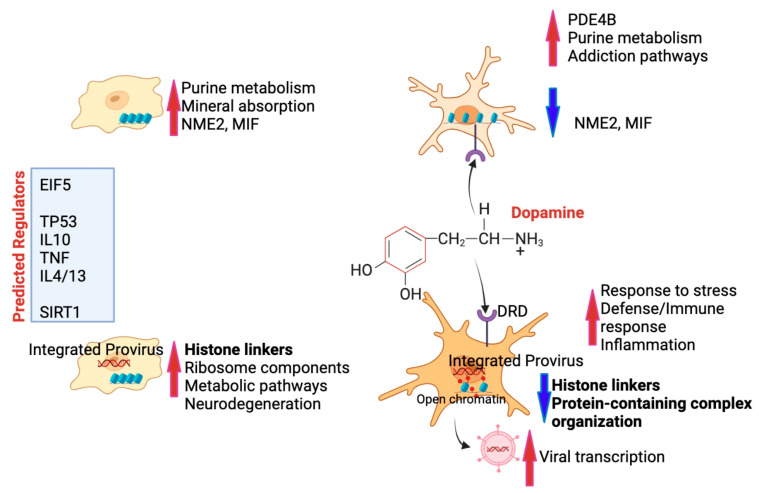
Summary of findings—Innate immune cells are maintained in a homeostatic resting state with the expression of strong signatures, such as NME2 and MIF, maintaining physiological processes via a healthy purine metabolism and mineral absorption (upper left). In cells that bear HIV provirus and are virologically suppressed, silence is maintained by a high expression of histone linkers and other components of chromatin organization, at the cost of a higher metabolic activation that may lead to some neurodegeneration (lower left). In the context of high levels of DA, innate immune cells do increase pathways associated with addiction (mainly dopaminergic signaling due to the presence of DA receptors in the innate immune cell surface) and are correlated with a decrease in the latency signature genes NME2 and MIF (upper right). On the other hand, cells with a provirus that are exposed to DA are strongly affected, with a significant decrease in histone linkers and protein-containing complex organization, unrepressing inflammation and unleashing HIV transcription. Predicted regulators of these phenotypes include EIF5, TP53, immune cytokines such as TNF, IL10, IL4 and IL13, as well as epigenetic silencers such as SIRT1.

**Table 1 viruses-17-00895-t001:** Cell Ranger-derived quality assessment and count summary.

	Estimated Number of Cells	Total Genes Detected	Median UMI Count per Cell	Median Genes per Cell
U1 NT	11,322	15,430	12,664	3349
U1 DA	8706	15,222	4702	1875
U937 NT	8721	24,571	8852	3106
U937 DA	11,933	24,866	7518	2858

## Data Availability

Data supporting reported results can be found at the Gene Expression Omnibus query dataset GSE278043.

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
